# Analysis of Microbial Diversity and Metabolites in Sauerkraut Products with and without Microorganism Addition

**DOI:** 10.3390/foods12061164

**Published:** 2023-03-09

**Authors:** Yueyi Liu, Xiaochun Chen, Fuxiang Li, Huiling Shi, Mingyi He, Jingping Ge, Hongzhi Ling, Keke Cheng

**Affiliations:** 1Engineering Research Center of Agricultural Microbiology Technology, Ministry of Education & Heilongjiang Provincial Key Laboratory of Plant Genetic Engineering and Biological Fermentation Engineering for Cold Region & Key Laboratory of Microbiology, College of Heilongjiang Province & School of Life Sciences, Heilongjiang University, Harbin 150080, China; liuyueyi2022@163.com (Y.L.); shl3589@126.com (H.S.); hhmmyyshiwo@163.com (M.H.); gejingping@126.com (J.G.); 2Engineering Research Center of Health Food Design & Nutrition Regulation, School of Chemical Engineering and Energy Technology, Dongguan University of Technology, Dongguan 523808, China; c_xiaochun95@163.com (X.C.); fuxianglee93@163.com (F.L.); 3Dongguan Institute of Technology Innovation, Dongguan 523000, China

**Keywords:** northeast sauerkraut, microbial composition, metabolome, correlation analysis

## Abstract

The microbial compositions and metabolites of fermented sauerkraut with and without the addition of microorganisms have been compared. The OTU clustering, nonvolatile compounds, volatile compounds and associations between bacterial taxa and metabolites were analyzed by 16S rRNA high-throughput sequencing technology, ultra performance liquid chromatography (UPLC), gas chromatography ion mobility mass spectrometry (GC-IMS) and the O2PLS model studies. The results showed that at the phylum level, the microbial species in the four sauerkraut types consisted mainly of the phyla *Firmicutes* and *Proteobacteria*, but different modes of microbial addition formed their own unique microbial communities. There were significant differences in the microbial communities among different northeast China sauerkraut samples, and different microbial communities exerted similar effects to inhibit *Firmicutes* production. At the genus level, sauerkraut without added microorganisms had the lowest microbial diversity. A total of 26 amino acids and 11 organic acids were identified and were more abundant in nonmicrobially fermented sauerkraut; 88 volatile organic compounds were identified in the 4 types of sauerkraut, with the microbially fermented sauerkraut being richer in alcohols, esters and acids. Different brands of sauerkraut contain their own unique flavor compounds. Cystine and tyrosine, ascorbic acid and acetic acid, and alcohols and esters are closely related to a wide range of microorganisms in sauerkraut. Elucidating the correlations among microbiota and metabolites will help guide future improvements in sauerkraut fermentation processes.

## 1. Introduction

Sauerkraut is a pickled lactic acid fermentation product made from cabbage with low salt concentrations and is widely consumed in northeastern China [1]. Sauerkraut contains high levels of probiotics, vitamins and minerals and has many advantages [2], such as the promotion of weight loss, antioxidation, and anticancer activity and lowering cholesterol, making it a beneficial health food [3,4]. In the 1950s, Pederson [5] studied the microorganisms in the vegetable fermentation process. For the first time, he pointed out that *Leuconostoc mesenteroides* initiated vegetable fermentation, and Liang et al. [6] further confirmed this result. Therefore, lactic acid bacteria (LAB) play a vital role in sauerkraut fermentation. However, traditional fermented sauerkraut is limited to a technical level, which leads to a relatively simple taste, poor product quality and unclear health function [7,8]. To meet the high-level demand for abundantly nutritious, health-promoting and flavor-rich foods, a variety of different flavor sauerkraut brands began to appear in the market [9,10].

Flavor is an important component in quality assessments of sauerkraut and mainly includes sour, fresh and savory flavors [11]. During sauerkraut fermentation, flavor characteristics are improved by metabolites produced by microorganisms, including amino acids, organic acids and volatile organic compounds [12]. Different fermentation agents and fermentation methods can lead to different flavor profiles. Yang et al. [13] investigated the quality characteristics of sauerkraut fermented by native lactic acid bacteria using sauerkraut from northeastern China as a raw material. The results showed that there were significant differences in the volatile profiles of sauerkraut from different fermentation agents, with ethyl lactate, ethyl acetate and isoamyl acetate being highest in sauerkraut inoculated with *L. plantarum*. Hu et al. [14] studied the microorganisms and metabolites in mixed sauerkraut fermented by any two of the four indigenous lactic acid bacteria and showed significant increases in the numbers of LAB and flavor components in mixed fermented sauerkraut, particularly with regard to the formation of leucine flavor. The results described above suggest that different fermenters develop different flavor characteristics. However, there are few studies on the effect of either single-strain or mixed-strain fermentation on the quality of sauerkraut.

Therefore, the purpose of this study was to explore in depth the effects of adding or not adding microorganisms on the properties of northeast Chinese sauerkraut and to compare the microbial compositions, nonvolatile compounds and volatile compounds of different sauerkraut varieties. This study will provide a reference to identify northeast Chinese sauerkraut brands. The research results are conducive to monitoring the sauerkraut process and establishing quality standards and have practical significance for improving the quality stability of sauerkraut and improving industrial development levels.

## 2. Materials and Methods

### 2.1. Selection of Sauerkraut Samples

The materials used for the experiments and microorganisms added are shown in Table 1. All sauerkraut samples were purchased from Harbin, Heilongjiang, China. *L. paracasei* was added to LLS sauerkraut, *L. plantarum* was added to HX sauerkraut, *L. plantarum* and *L. acidophilus* were added to QM sauerkraut. ZLL sauerkraut was naturally fermented without a starter culture. Moreover, three kinds of sauerkraut with added microorganisms are currently the most popular types in northeastern consumer markets, and sauerkraut liquids were selected as the study samples and stored in an −80 °C refrigerator (Hoshizaki Electric Co., Ltd., Nagoya, Japan) for further analysis [13].

### 2.2. Physicochemical Analysis

The pH levels were determined using the pH electrode of a pH meter (Mettler-Toledo Co., Ltd., Zurich, Switzerland) placed directly into each sample. Total acids in sauerkraut were determined by following the method of Pino et al. [15]. Total sugars were determined through the 3, 5-Dinitrosalicylic acid (DNS) method using a spectrophotometer (Mettler-Toledo Co., Ltd., Zurich, Switzerland). The salt contents of the four sauerkraut types were determined by the Mohr method [1].

### 2.3. Bioinformatics Analysis

The sauerkraut liquids of four kinds of sauerkraut were selected as study samples for microbial diversity identification. Three parallel samples for each sample were sent to Shanghai Majorbio Company for 16S rRNA high-throughput sequencing. Small-fragment gene libraries were constructed for single-end sequencing. The amplified region was the V3-V4 region of the 16S rRNA gene with 338F (5′-ACTCCTACGGGAGGCAGCAG-3′upstream and 806R (5′-GGACTACHVGGGTWTCTAAT-3′downstream. Additionally, the sequences obtained were analyzed for quality control using the online tool of Fastp (Haplox Co., Ltd., Shenzhen, China). OTU clustering analysis was performed at a 97% similarity level, which in turn was used for taxonomic analysis using the RDP classifier Bayesian algorithm [16]. Measures of alpha diversity, including Shannon and Simpson species diversity indices and Ace and Chao1 species richness estimators, were calculated using QIIME (1.9.1). Beta diversity analysis was performed with two indicators, weighted UniFrac and unweighted UniFrac distances, to determine the coefficients of dissimilarity among the samples. LEfSe software was used to identify those communities or species that generated significant levels of dissimilarity among samples, and the linear discriminant analysis score was set to 4. After standardization of the OTU data, the top 30 genera in terms of abundance were selected for univariate network analysis, and network diagrams were drawn based on species-to-species correlations and were used to reflect the interactions among species in the samples [17,18].

### 2.4. Determination of Metabolite Groups

#### 2.4.1. Determination of Nonvolatile Compounds

The levels of nonvolatile compounds (e.g., organic acids and amino acids) were determined using a UPLC-Q-TOF/MS (AB Sciex Co., Ltd., Framingham, MA, USA) instrument, following the method described in detail by Kang et al. [19] with slight modifications. Sauerkraut juice was centrifuged at 1948 g for 10 min and then filtered through a 0.22 µm microporous filter before being sampled on an Agilent HC-C18 (4.6 mm × 150 mm, 5 µm) column with mobile phases of 0.1% formic acid in water (solvent A) and acetonitrile (solvent B).

Amino acids were analyzed in positive ion scanning mode using an injection volume of 3 μL, flow rate of 0.3 mL/min, column temperature of 35 °C and gradient elution program consisting of 0–2 min, 99% A; 2–3.25 min, 99–95% A; 3.25–4.25 min, 95% A; 4.25–7.75 min, 95–45% A; 7.75–9.75 min, 45–10% A; 9.75–11.75 min, 10% A; 11.75–12 min, 10–99% A; and 12–15 min, 99% A. The other parameters were set as follows: electrospray pressure, 3.5 kV; capillary temperature, 320 °C; acquisition range, 50–1000 *m/z*; resolution, 70,000; sheath gas, 40 Arb; curtain gas, 10 Arb; and protective gas heating temperature, 350 °C [20].

Organic acid detections were performed in negative ion scanning mode with an injection volume of 5 μL, flow rate of 0.25 mL/min and column temperature of 50 °C. The injection conditions were as follows: 0–2 min, 99% A; 2–3.25 min, 99–95% A; 3.25–4.25 min, 95% A; 4.25–7.75 min, 95–45% A; 7.75–9.75 min, 45–10% A; 9.75–11.75 min, 10% A; 11.75–12 min, 10–99% A; 12–15 min, 99% A; and spray pressure, 4.5 kV.

#### 2.4.2. Determination of Volatile Compounds by GC-IMS

HS-GC-IMS (Gesellschaft für Analytische Sensorsysteme mBH, Dortmund, Germany) was used to detect and analyze the volatile metabolites. A slight modification of one of the analytical methods of Yang et al. [13] was used. The following method was used: 2 mL of sample was placed in a 20 mL headspace vial, incubated at 3× *g* for 15 min at 60 °C and then injected into the sample with an injection needle at 85 °C and no splitting. Volatile compounds were separated using an MXT-WAX-1 organic column (15 m; ID: 0.53 mm; analysis time, 26 min; injection volume, 500 μL; and drift gas flow (N_2_) set at 150 mL/min) (Shimadzu Research Laboratory Co., Ltd., Shanghai, China). The elution conditions were as follows: the flow rate was held at 2 mL/min for 2 min, increased to 10 mL/min from 2–10 min, increased to 100 mL/min from 10–20 min and increased to 150 mL/min from 20–25 min. GC-IMS analyses were performed in triplicate. Retention indices (RIs) were calculated using a mixture of n-ketones (C4–C9). Volatile compounds were characterized according to RI and drift time (DT) using the NIST database (https://kinetics.nist.gov/kinetics/ (accessed on 17 October 2022)) and GC x IMS library search software that is included with the IMS database (https://www.ims-gmbh.de/ (accessed on 17 October 2022)).

### 2.5. Data Analysis

To analyze the links between bacteria and metabolites, O2PLS analysis was performed with SIMCA 14.1 (Sartorius Stedim Biotech Co., Ltd., Gottingen, Germany) [21] by using projections to analyze the significances of the relationships among bacteria and metabolites, which was further validated using Spearman correlation analysis.

According to the least significant difference test (LSD) method, all experiments were performed with at least three biological replicates. The data represent the means plus or minus the standard deviations (±SD). Differences were considered statistically significant when the *p* values were less than 0.05.

## 3. Results and Analysis

### 3.1. Basic Physicochemical Properties

The basic physicochemical properties of the four sauerkraut types are shown in Table 2. pH was used to evaluate sauerkraut maturity and is an important factor affecting the microbial community during sauerkraut fermentation [20]. The pH differences among the four sauerkraut types were not significant, with HX sauerkraut having the lowest pH (3.59) and ZLL sauerkraut having the highest pH (3.80). The pH values of the sauerkraut with added microorganisms were all lower, which may be related to the homologous fermentation properties of *L. plantarum* and *L. paracasei* [22]. Low pH levels limit the growth of spoilage bacteria. Some studies have shown that the pH of finished sauerkraut is in the range of 3.0–4.0 [23].

Total acid contents were inversely proportional to pH contents. HX and LLS sauerkraut exhibited higher total acid contents than the other two fermentation methods. Because different varieties of sauerkraut use different raw materials and the fermentation temperatures, times and dominant fermentation of bacteria and fermentation process are also different, different organic acid contents are generated, resulting in different total acid contents. Elevated total acid contents significantly accelerated the initiation of fermentation, indicating a higher souring performance of sauerkraut fermented with a single bacterial species. A shorter acidification process leads to faster conversion of sauerkraut, fewer commercial losses and lower production costs [24].

Reducing sugars are the main carbohydrates converted into lactic acid and sauerkraut flavor and aroma [25]. The experimental results showed differences in the levels of reducing sugars. The fermentation systems of sauerkraut are different, the kinds and numbers of lactic acid bacteria differ, and lactic acid bacteria and other microorganisms in sauerkraut use the reduced sugars in sauerkraut to participate in their own metabolism for growth and reproduction. Starches in sauerkraut can also be hydrolyzed to produce certain amounts of reducing sugars, so the contents of reduced sugars will be different. The lower contents of reducing sugars in HX and QM sauerkraut may be related to the high digestion of reducing sugars by microorganisms such as lactic acid bacteria and yeast [26].

Salt content is a key parameter in sauerkraut fermentation and storage. Adequate salt contents inhibit the growth of spoilage bacteria, and in the early stages of fermentation, salt draws nutrients from raw cabbage and provides a substrate for microbial growth [27]. Salt combines with the acids produced by fermentation to inhibit growth of microorganisms that are not important in sauerkraut fermentation and delays the enzymatic softening of sauerkraut [28]. QM sauerkraut had the lowest salt content, and the remaining sauerkrauts did not differ significantly in their salt contents.

### 3.2. Microbiological Analysis

#### 3.2.1. Microbial Statistical Analysis

A total of 605,269 high-quality gene sequence datasets were generated from the 4 sauerkraut types, with an average of 428 sequences per sample (range from 421 to 440). A total of 428 OTUs were obtained at a 97% similarity level. The products showed 100% coverage in all samples, indicating that the number and quality of the sequences were sufficient to reveal most of the microbial OTUs in the samples [29].

#### 3.2.2. Alpha Diversity Analysis

The QIIME platform was used to calculate alpha diversities and to evaluate the abundances of taxa and diversity of microbial communities. The Shannon and Simpson indices were used to evaluate the diversities of microbial communities. The lower the Simpson index, the higher the microbial diversity, while the opposite is true for the Shannon index [30]. In this study, LLS had the highest community diversity, the diversity of HX was not significantly different from that of LLS, and QM and ZLL had similar bacterial diversities (Figure 1A). The Chao1 and ACE richness indices further validated this result (Figure 1B). The bacterial diversity results were consistent with the pH results, since differences in pH affect differences in microbial diversity. During fermentation, greater numbers of exogenous microorganisms that cannot tolerate a highly acidic environment are temporarily inhibited due to the accumulation of acid, and as the microorganisms adapt to the acidic and saline environment, the microbial diversity subsequently rises [31].

#### 3.2.3. Structure of Bacterial Populations

The samples were subjected to OUT cluster analysis. The number of OTUs shared by the three sauerkraut samples was 370: 168 for LLS, 293 for HX, 89 for ZLL and 76 for QM. The results indicated that the community diversity and richness of sauerkraut with starter culture were higher than those of naturally fermented sauerkraut during storage.

A total of 22 phyla and 228 genera were identified in the microbial communities, and their relative abundances are shown in Figure 1C,D. The top five most abundant phyla among the four sauerkraut types were *Firmicutes*, *Proteobacteria*, *Cyanobacteria*, *Bacteroidota*, and *Actinobacteriota*, but other phyla with lower relative abundances were also detected. *Firmicutes* and *Proteobacteria* can decompose the sugar in sauerkraut during the fermentation process to produce ethanol and acetic acid, improving its flavor. In addition, some beneficial enzymes such as proteases, glycases, and amylase are produced to accelerate the fermentation process of sauerkraut [32]. *Cyanobacteria* can inhibit the production of spoilage bacteria during the acid fermentation process, improve the quality of sauerkraut, and extend the shelf life [33]. During storage, *Firmicutes* had the highest abundance, with a relative abundance of 83.3–99.9%. Its relative abundance in naturally fermented sauerkraut is higher than that in fermented sauerkraut with starter cultures. The abundance of *Firmicutes* at 45 d of storage was significantly higher than at the beginning of storage, indicating that the storage environment changed the composition of the bacterial community. At the beginning of storage, the proportions of *Firmicutes* in the LLS, HX and ZLL groups were 83.3%, 90.3% and 99.7%, respectively, indicating that different strains have similar effects, which inhibit *Firmicutes* production, and the inhibitory effects of the two different strains on *Firmicutes* production would be different. At 45 d, the *Firmicutes* proportions in the three groups were 99.5%, 98.7% and 99.9%, respectively. The LLS group exhibited the most significant change, indicating that the inhibition was weakened and the *Firmicutes* proportions gradually increased during storage. At the genus level, there were significant differences in the dominant genera among the LLS and other samples. The abundant genera in the LLS samples consisted of *Lactobacillus* (73%), *Chlorobacillus* (10%), *Lactobacillus* (4.3%) and *Lactococcus* (3.3%). The abundances of genera in the HX samples were not significantly different from those in the LLS samples. The compositions of the HX and LLS samples were significantly different from those of the QM and ZLL samples; the QM and ZLL microbiotas were less diverse, with *Lactobacillus* and *Proteobacteria* dominating. *Lactobacillus* can produce a variety of antimicrobial substances, including lactic acid and bacteriocins, which inhibit pathogen growth and spoilage microorganisms and therefore reduce the proportions of other genera, which is also consistent with the pH and TA values measured in sauerkraut with *Lactobacillus*. Yang et al. [13] showed that natural fermentation leads to sharp reductions in *Proteobacteria*. Zhou et al. [34] showed that *Lactobacillus* dominated in the later stages of fermentation, and *Lactobacillus* and *Bacteroides* produced carbohydrate-active enzymes that promoted carbohydrate hydrolysis. Zhu et al. [31] showed that the microbial diversity of sauerkraut depends not only on the fermentation process but also on the fermenting agent and fermentation environment.

#### 3.2.4. Analysis of Microbial Beta Diversity

Appendix A show heatmaps of the beta diversity indices of different sauerkraut samples. In the beta diversity analysis, two indicators, weighted UniFrac and unweighted UniFrac distances, were chosen to measure the coefficients of variation between two samples. The small β values for QM, ZLL and HX sauerkraut indicate that there is very little difference among these three sauerkraut types, and the values representing variability between LLS sauerkraut and the other three types are larger.

The differences among the microbial communities of the different types of fermented sauerkraut were compared by principal coordinate analysis (Appendix A). The two eigenvalues of the principal component analysis were 65.53% (PC1) and 22.65% (PC2), and the scatterplot showed that the different fermented sauerkraut samples were automatically clustered into four groups according to their OTU compositions. All four groups of sauerkraut were significantly different in the middle, and the four groups were located in four quadrants with significant separations among groups. The results indicate that the microbial communities in northeast Chinese sauerkraut vary with different fermenting agents. Different modes of microbial addition formed their own unique microbial communities, and there were significant differences in the microbial communities among different northeast Chinese sauerkraut samples.

#### 3.2.5. Network Characteristics of Sauerkraut Microorganisms

Through a correlation analysis of the network diagram, the species abundance information of different samples was studied, the coexistence relationships of species in the environmental samples were obtained, and the properties of the microbial networks of the four sauerkraut types were compared and analyzed, which highlighted the similarities and differences among the samples. The top 30 genera in terms of abundance were selected for network analysis with ρ ≥ 07. The results are shown in Figure 1E, with a network transmissibility of 0.83, network diameter of 4 and average shortest path length between nodes of 1.75. The network was connected, and there was a tendency for the microorganisms in sauerkraut to form symbioses.

Genus-level correlation network plots revealed significant interrelationships among different genera in the four sauerkraut species, with red lines indicating positive correlations and blue lines indicating negative correlations and with the colors of the connecting lines indicating the magnitudes of the correlation coefficients and the same node colors indicating the same phylum. Among the 30 nodes, there were 8 types of bacterial phyla, of which the phyla *Firmicutes* and *Proteobacteria* were widely distributed, and 14 nodes had the highest connectivity in the network diagram, which were related to 17 genera. Their cluster coefficients were above 0.88, indicating high correlations with other genera. In addition, the highest clustering coefficient was for *Sphingomonas*, which belongs to *Proteobacteria*.

#### 3.2.6. Discovery of Biomarkers in Sauerkraut

In this paper, LEfSe was used to distinguish the biological taxa in the four types of sauerkraut, and the nonparametric factorial Kruskal–Wallis (KW) rank sum test was used to detect differential traits with significant abundances, find taxa with significant differences in abundance, and use linear discriminant analysis (LDA) to estimate the contributions of species abundances to the differences. The distribution of biomarkers for sauerkraut in the branching diagram according to the phylogeny is shown in Figure 2A. A total of six biomarkers were identified for the four sauerkraut types, with all biomarkers occurring at the genus level in all samples except for QM sauerkraut, where no biomarkers were identified. *Leuconostoc*, *norank_f__norank_o__Chloroplast* and *Lactococcus* were biomarkers for LLS. *Kosakonia* and *Pediococcus* were biomarkers for HX. *Lactobacillus* was a biomarker for ZLL.

To further validate the results described above, taxa with LDA scores greater than 4 were screened as biomarkers (Figure 2B). The four sauerkraut types differed in the microbial markers obtained due to the different fermenting agents used, but the biomarkers in all four sauerkraut types were microorganisms indigenous to the sauerkraut fermentation process. The results described above indicate that sauerkraut is a continuously fermented food and that LABs are the most dominant microorganisms in the vegetable fermentation process [6].

### 3.3. Analysis of Targeted Metabolomic Profiles

#### 3.3.1. Detection and Analysis of Amino Acids in Sauerkraut

LAB can catabolize proteins into amino acids, thereby improving the flavor and nutritional value of northeastern sauerkraut. FAA not only has a unique flavor but can also participate in the biosynthesis of flavor as a direct precursor to produce a variety of aroma metabolites, such as acids, alcohols, esters and carbonyl groups, in fermented foods. In this study, a total of 26 amino acids were detected, the types and relative contents of which varied among sauerkraut samples (Appendix A). This result may be related to the proteolytic activities associated with the microbial enzymes in different starter cultures, suggesting that different starter cultures may alter sauerkraut flavor by regulating amino acid metabolism. The highest amino acid content was found in ZLL sauerkraut, followed by QM sauerkraut. The amino acid compositions were most abundant in phenylalanine, followed by leucine, isoleucine, tryptophan, histidine and proline. Similar results have been reported in previous studies on sauerkraut fermentation, and Abubakr et al. [35] demonstrated better proteolytic activities of *L. plantarum* and *Lactobacillus entericus*, with *Lactobacillus plantarum* being more active than *Lactobacillus entericus*. *Streptococcus paracasei* had the highest protein hydrolytic power compared to short-chain *Lactococcus* and *Lactobacillus plantarum* isolated from conventional cheese.

To explore the effects of different fermentation techniques on amino acids more visually, a score plot of the amino acid compositions of different sauerkraut types based on the O2PLS-DA model was constructed, and the results are shown in Appendix A. The score plot shows that the contents and types of amino acids can effectively separate the four types of sauerkraut, among which ZLL, QM and LLS are listed in order from left to right, indicating that the amino acid compositions changed significantly with the application of fermenting agents, and HX sauerkraut was significantly distant from the other three types, indicating that different fermenting agents produced differences in amino acid compositions.

A biplot of the amino acids of the different types of sauerkraut based on the O2PLS-DA model is shown in Figure 3A. ZLL sauerkraut was positively associated with a number of amino acids, including ornithine, o-phosphoserine and 1-methylhistidine. Serine has a sweet taste and can subsequently be converted by deamidation reactions to acetic acid. 2,3-Serine was positively associated with ZLL. Serine was not detected in QM sauerkraut and its levels were not significantly different between LLS and HX sauerkraut, suggesting that sauerkraut fermentation with added bacteria affects serine production [14]. LLS sauerkraut was also positively associated with cystine, alanine, aminobutyric acid and hydroxylysine. Some studies [13] show that alanine is sweet and aminobutyric acid is an amino acid derivative generated by the irreversible decarboxylation of glutamic acid in LAB by the enzyme glutamic acid decarboxylase, causing a hypotensive effect in the central nervous system for its basal role and is studied with respect to tubular valves, where it is involved in regulating blood pressure and heart rate and relief from pain and anxiety [3]. Cystine contents depend on the hydrolytic activities of endogenous and microbial enzymes on myosin and myofibrin and make an important contribution to improving the nutrition of the product. HX sauerkraut is positively associated with glutamine, and QM sauerkraut contains no obvious amino acid markers.

#### 3.3.2. Detection and Analysis of Organic Acids in Sauerkraut

Probiotics can degrade reducing sugars through fermentation, leading to the formation of organic acids, which constitute an important secondary carbon source that allows many microorganisms to proliferate during fermentation [36]. In this study, 11 organic acids were detected in the four different types of sauerkraut (Appendix A). In previous studies, lactic acid has been found to play a dominant role in the formation of sourness in sauerkraut, with citric, acetic, oxalic, malic, succinic and tartaric acids playing secondary roles, and appropriate levels and proportions of these organic acids contribute to the gentle sourness of sauerkraut. Appendix A shows that there were no significant differences in the tartaric, succinic and ascorbic acid contents of the four fermented sauerkraut types. The lactic acid contents were significantly higher in sauerkraut fermented with mixed starter culture strains than in sauerkraut fermented with a single strain and naturally fermented sauerkraut. Lactic acid is the main organic acid in sauerkraut and gives it a softer taste and replaces acidity and astringency while improving the microbial stability of fermentation products. The variations in lactic acid contents indicate that the lactic acid concentrations produced by LAB are related to the strain. The citric acid contents in the two sauerkrauts fermented with a single strain were higher than those of mixed-strain fermented sauerkraut and naturally fermented sauerkraut. Citric acid is found in many substrates used for food fermentation and is metabolized into flavor compounds, such as diacetyl, ethyl acetate, butylene glycol and acetaldehyde. Smid et al. [37] described the conversion of the citric acid degradation pathway to an amino acid transamination pathway, potentially leading to the formation of aromatic compounds. The entire organic acid group was dominated by succinic acid and pyroglutamic acid, which accounted for 61.5% and 34.3% of the overall organic acid content, respectively, indicating that inoculation with lactic acid bacteria did not affect the overall organic acid ratios.

A score plot for the organic acid compositions of different sauerkraut types based on the O2PLS-DA model is shown in Appendix A. The score plot shows that the contents and types of organic acids can effectively separate the four types of sauerkraut, with LLS and QM sauerkraut containing similar organic acid compounds and ZLL sauerkraut being significantly distant from the remaining three types of sauerkraut, indicating that single- and mixed-strain fermentations promote organic acid production. A biplot for the organic acids in the different types of sauerkraut is shown in Figure 3B. ZLL sauerkraut was positively associated with tartaric acid, malic acid and lactic acid. LLS sauerkraut was positively associated with lactic acid, ascorbic acid and lactic acid. QM and HX showed no significant positive associations.

### 3.4. Analysis of Volatile Compounds

Volatile organic compounds (VOCs) are some of the most important indicators of the freshness and nutritional value of food and are essential for the sensory properties and acceptability of food. In this study, the volatiles were analyzed qualitatively and compared quantitatively using LAV software and the NIST and IMS databases. Potential markers were identified in a 2D plot (Appendix A) by reaction enhancement, encompassing retention time (*Y*-axis), DT (*X*-axis) and ion signal intensity. These results show that the volatile substances in different samples were similar for different fermentation agents and methods [38]. However, the peak signal intensities differed, indicating different concentrations of volatile compounds. The peak signal intensity data for the volatiles in the four sauerkraut types are shown in Appendix A. All measurements were repeated three times.

GC-IMS identified a total of 88 volatile compounds, some of which occurred as monomers and dimers due to their concentrations and properties. After qualitative analysis based on the database, the totals included 24 alcohols, 17 aldehydes, 11 esters, 9 ketones, 6 acids and 21 other compounds. The types of volatile flavor compounds in the four sauerkraut samples were generally consistent, but there were corresponding differences in the contents and concentrations of the different types of compounds in the different sauerkraut types due to factors such as raw materials, fermentation strains, fermentation temperatures and processing.

The characteristic flavor fingerprints of the four different sauerkraut types were constructed using the Gallery plot plug-in. The results are shown in Figure 4A–C. The results show that differences in strains and fermentation methods produced variations in volatile organic compounds in the sauerkraut samples. Some alcohols (e.g., propylene glycol, 1-propanol, and heptanol) and esters (e.g., methyl butyrate, methyl isovalerate, and butyl propionate) were not present in the ZLL samples. Alcohols, esters and acids were more abundant in sauerkraut fermented with added bacteria. The most diverse compounds in the four sauerkraut types were alcohols, with LLS sauerkraut having the greatest number of alcohols at 18. Alcohols tend to have distinctive flavor characteristics (aromatic and earthy), and they are probably the main contributors to the flavor of sauerkraut. Production of alcohols with aromatic flavors through microbial fermentation or amino acid degradation can improve the flavor of fermented foods [9]. Various C6-C10 alcohols and aldehydes are volatile organic compounds produced by nonmicrobial action [37]. Esters are often reported to have fruity aromas. Pyrazines, which are specific products of the Maillard reaction and have a distinctive flavor, were found in both monoculture and mixed-strain fermented sauerkrauts. The maximum number of pyrazine species in QM sauerkraut was three, and the pyrazine found in all samples was 2,5-dimethylpyrazine, which may be a result of the Maillard reaction occurring in sauerkraut during fermentation.

The four types of sauerkraut were subjected to PCA to compare and analyze the differences in volatile flavor among the sauerkraut types. The PCA of the different types of sauerkraut is shown in Figure 4D. The sum of the contributions of principal component 1 (PC1) and principal component 2 (PC2) was 78.1%, indicating significant differences among samples with different fermenting agents. The four sauerkraut samples were clearly separated from each other. LLS sauerkraut was located far from the other sauerkrauts, indicating that its volatile components were more different from those contained in the other sauerkrauts. The QM and HX sauerkraut samples were clustered on the right side of the image and were close together, indicating similar volatile components.

The biplot results (Appendix A) show that ZLL sauerkraut is located in the upper part of the image and that its characteristic volatile components were methylpyrazine, z-3-hexen-ol, heptanol, 2-octanol and 2-methylpropanoic acid. HX sauerkraut is located in the second quadrant, and its volatile components were α-pinene, phenylacetaldehyde and 2-cyclohexen-1-one. QM is located in the third quadrant and is characterized by the volatile compounds, N-diethylethanamine and benzaldehyde trans-2-heptenal. LLS sauerkraut is located in the fourth quadrant, and its characteristic volatile compounds were ethyl-2-methylpropanoate, 1-pentanol, E-2-octenal, 1,8-cineole, and isobutyl acetate.

### 3.5. Associations between Bacterial Taxa and Metabolites

To predict the potential associations among bacterial taxa and metabolites in sauerkraut, the O2PLS model was used. The model was further validated based on Spearman correlation analysis to explore whether there were strong correlations among bacterial taxa and metabolites.

#### 3.5.1. Analysis of Correlations among Microbial Taxa and Amino Acids

The correlations among amino acids and the top 30 bacterial taxa in terms of abundance are shown in Figure 5A. O2PLS generated a model with R^2^ = 0.99 and Q^2^ = 0.78, indicating that amino acids are closely related to microbial communities. Figure 5A shows that cystine, plusallo-σ-hydroxylysine, tyrosine, valine O-phosphoethanolamine, glutamine, asparagine, theanine, and arginine were significantly correlated with bacterial taxa. Appendix A further validated these results with correlations of cystine and tyrosine with 18 genera, indicating that these two amino acids are important in sauerkraut. The LLS samples were closely associated with cystine and tyrosine, suggesting that *L. paracasei* fermentation of sauerkraut is more conducive to the production of these two amino acids. In addition, *Lactobacillus* and *Pediococcus* were the main contributors to theanine and asparagine; *Weissella* was the main contributor to glutamine.

#### 3.5.2. Analysis of the Correlations among Microbial Taxa and Organic Acids

The correlations among organic acids and the top 30 bacterial taxa in terms of abundance are shown in Figure 5B. O2PLS generated a model with R^2^ = 0.99 and Q^2^ = 0.78, indicating that organic acids were closely correlated with microbial taxa. Figure 5B shows that oxalic acid, ascorbic acid, citric acid, lactic acid, fumaric acid, butanedioic acid and pyruvic acid were significantly correlated with floral taxa, while the remaining organic acids were not significantly correlated with any taxa. The flora were mostly clustered on the right side of the image, closer to oxalic acid, while ascorbic acid and lactic acid were closely correlated with LLS. Appendix A further verifies the results described above. Ascorbic acid and acetic acid were the important organic acids and were significantly correlated with the 13 bacterial taxa. Acetic acid has a sour odor and makes a significant contribution to sauerkraut flavor.

#### 3.5.3. Analysis of Correlations among Microbiota and Volatile Compounds

Currently, there are few studies on the correlations among microflora and volatile metabolites in sauerkraut. In this study, the Spearman correlation analysis method was used to integrate the microbiota and volatile group data for northeastern Chinese sauerkrauts, and the potential correlations among the microbiomes and volatile substances were explored. *Lactobacillus*, *Pediococcus* and *Lactococcus* showed strong correlations with alcohols and esters. The positive correlation between *Clostridium-sensu-stricto-*13 and ketones is related to the ability of laboratories to produce free fatty acids, which are precursors of flavorful substances (Figure 5C). The results of Appendix A show that alcohol volatiles were closely correlated with 15 genera. Analyses of the correlations of microflora and volatile metabolites contribute to the screening of starter cultures and optimization of fermentation processes to produce sauerkraut products with better flavor and higher nutritional value [38].

## 4. Discussion

The basic physicochemical properties, microbiota and metabolite differences in sauerkraut fermented by different fermenting agents were investigated. The results showed that different modes of microbial addition will form their own unique microbial communities, and different microbial communities have similar effects, which inhibit the production of *Firmicutes.* The single-strain fermented sauerkraut types, LLS and HX, exhibited higher microbial diversities. The contents of free amino acids in bacterially fermented sauerkraut were higher than those in naturally fermented sauerkraut, and there were nonsignificant differences in the organic acid contents. In general, the physical and chemical characteristics of added starter culture sauerkraut are better than those of naturally fermented sauerkraut products, and this conclusion is consistent with the results of Zhao et al. [39]. GC-IMS analysis revealed a richer variety of alcohols, esters and acids in the sauerkrauts fermented with added bacteria. The characteristic volatile components of ZLL sauerkraut include methylpyrazine, z-3-hexen-ol, heptanol, 2-octanol and 2-methylpropanoic acid. Pyrazines, which are specific products of the Maillard reaction, have a distinctive flavor. The characteristic volatile components of HX sauerkraut include α-pinene, phenylacetaldehyde and 2-cyclohexen-1-one. The characteristic volatile components of QM sauerkraut include N-diethylethanamine and benzaldehyde trans-2-heptenal. The characteristic volatile components of LLS sauerkraut include ethyl-2-methylpropanoate, 1-pentanol, E-2-octenal, 1,8-cineole and isobutyl acetate, and ethyl-2-methylpropanoate has flower and fruit fragrances. Alcohols give LLS sauerkraut the aroma of wine. The volatile compounds contributing significantly to flavor in addition-fermented sauerkraut were higher than those in naturally fermented sauerkraut. The correlations among different sauerkraut core taxa and metabolites were predicted using O2PLS, and the relationships among bacteria and metabolites were established under different fermentation agent conditions. This study provides a theoretical basis to improve the quality of sauerkraut and provides a reference for the identification of northeast sauerkraut brands. In addition, this study helps to establish sauerkraut brands.

## Figures and Tables

**Figure 1 foods-12-01164-f001:**
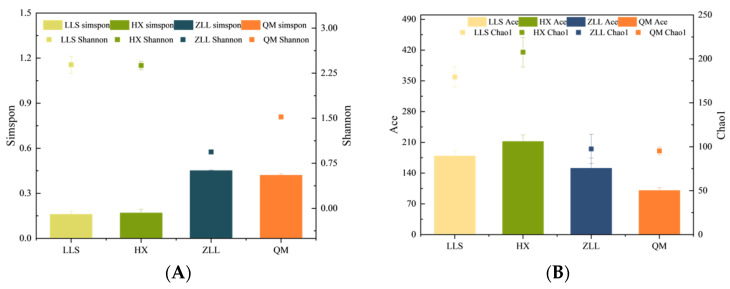
Microbial diversity analysis of sauerkraut. (**A**) Shannon and Simpson indices, (**B**) Chao1 and ACE indices, (**C**) phylum-level microbial composition, (**D**) genus-level microbial composition, (**E**) network analysis.

**Figure 2 foods-12-01164-f002:**
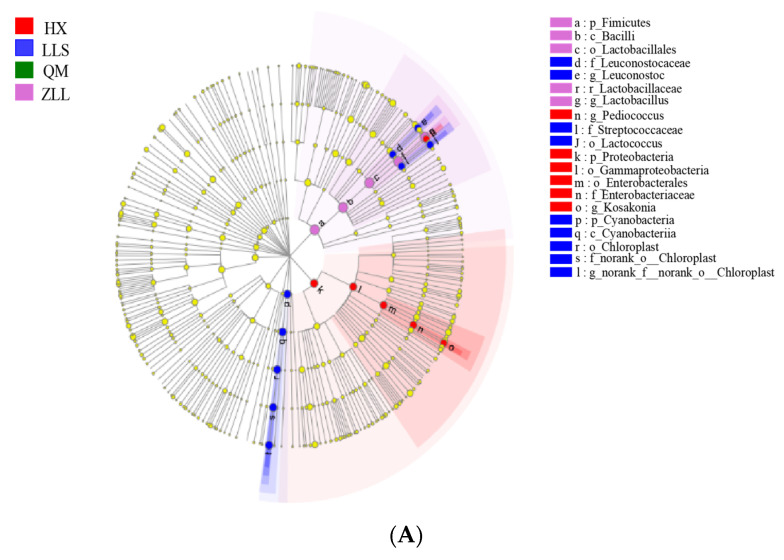
Analysis of biomarkers in sauerkraut. Branching plots (**A**) and histograms of LDA values (**B**) are used for labelling.

**Figure 3 foods-12-01164-f003:**
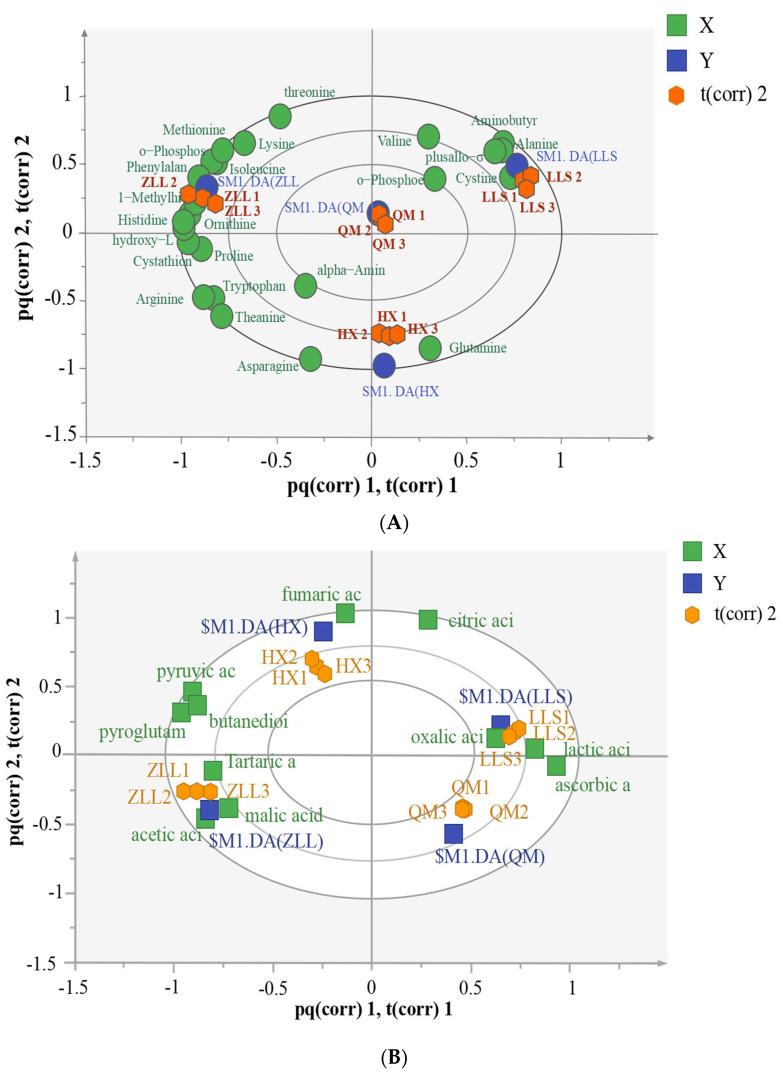
Targeted metabolome analysis. (**A**) amino acid biplot scores, (**B**) organic acid biplot score.

**Figure 4 foods-12-01164-f004:**
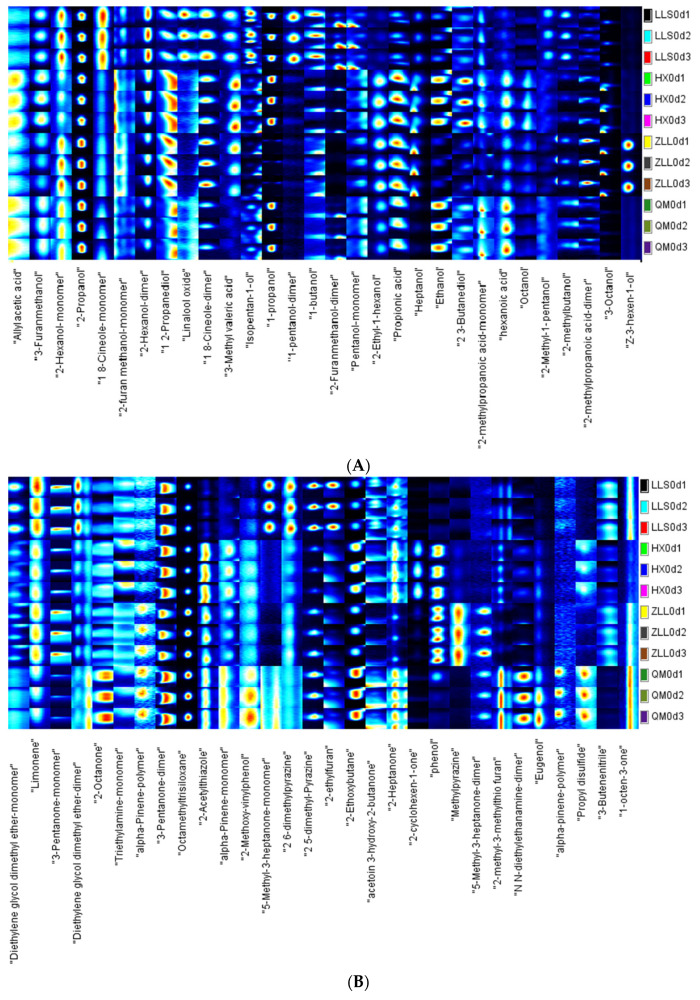
Analysis of volatile compounds. (**A**) Fingerprint mapping of acids and alcohols, (**B**) fingerprint mapping of ketones and others, (**C**) fingerprint mapping of aldehydes and esters, (**D**) PCA.

**Figure 5 foods-12-01164-f005:**
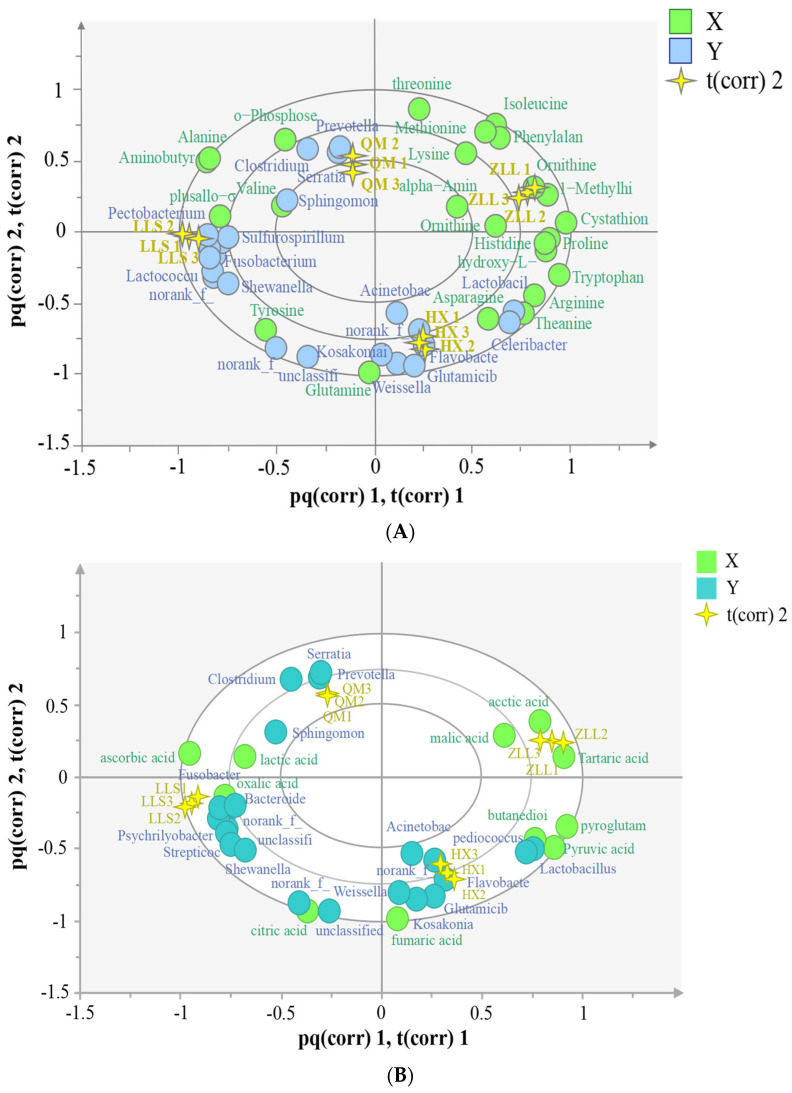
Correlations between bacterial taxa and metabolites. (**A**) O2PLS analysis of amino acids and bacterial taxa, (**B**) O2PLS analysis of organic acids and bacterial taxa, (**C**) O2PLS analysis of VOCs and bacterial taxa.

**Table 1 foods-12-01164-t001:** Starter cultures compositions of sauerkraut samples.

Sample	*L. paracasei*	*L. plantarum*	*L. acidophilus*
LLS	+	—	—
HX	—	+	—
QM	—	+	+
ZLL	—	—	—

**Table 2 foods-12-01164-t002:** Physicochemical properties of sauerkraut.

Physical and Chemical Properties	LLS	HX	QM	ZLL
pH	3.67	3.59	3.62	3.80
Total acid (g/kg)	10.98	14.06	10.44	9.00
Reduced sugar (g/100 g)	3.254	1.009	0.781	3.030
Salts (g/100 g)	1.40	1.87	0.94	1.87

## Data Availability

The data presented in this study are available on request from the corresponding author. All the data by the author can be obtained from the corresponding author upon request.

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
