# Peer review of "Analysis of Microbial Diversity and Metabolites in Sauerkraut Products with and without Microorganism Addition"

_foods, 2023, doi:10.3390/foods12061164_

Round 1

Reviewer 1 Report

The authors conducted a study on the composition of microbial populations and their metabolites of fermented sauerkraut with and without the addition of starter cultures. The manuscript is not well written and difficult to read and understand. The Materials and Methods section is so unclear and incomplete that it is difficult to understand the results. The results are insufficient and incomplete for relevant conclusions. The topic is interesting but not innovative as there are several similar papers in the literature, so I do not think this paper is suitable for further processing in Foods.

 COMMENTS
Introduction

The introduction is unclear and very confusing. The authors illustrate the role and importance of sauerkraut to the Chinese population by emphasising the introduction of starter cultures into the industrial product. In addition, various papers on sauerkraut inoculated with LAB starter cultures and bacterial fermentation products have been reported in a haphazard and unclear manner that makes it difficult to understand the text. The introduction needs to be more clearly stated and consistent with the objective of the paper.
Materials and methods

Methods must be justified by bibliographic references.

Methods modified by the authors must be briefly described.

Manufacturer and country of production must be given for each instrument.

The entire section on DNA extraction, PCR amplification and sequencing date is missing.
It is not clear how you identified the different types of LAB starter cultures in Table 1.

 Results and analysis

From the results of the physicochemical properties, it appears that the two samples studied, HX and LLS, had higher total acid contents. According to the authors, the acid production is influenced by the different production methods. Why did the authors not describe the different types of sauerkraut and the different production methods in the Materials and Methods section? The authors should have indicated the different fermentation methods used in the production of sauerkraut to better understand the role of LAB, which is influenced by the production process. This would have been useful to make considerations, otherwise they are just speculation.

How many OTUs were found per sample? The authors do not specify, please clarify.

Please replace “ addition of bacteria” with “starter cultures” throughout the text.

The incidence of the different phyla during sauerkraut ripening is not clear.

The incidence of the phyla Firmicutes during storage is not clear.

Which strains cause inhibition?

What other phyla were identified and in what percentages?

Have you considered storage times? It is unclear.

 Author Response

Reviewer 2 Report

In this manuscript Liu et al investigated four sauerkrauts, LLX, HX, QM and ZLL consumed in northeastern China and determined their physicochemical properties and metabolite and microbial composition. The authors performed different microbial diversity analysis to show that Firmicutes, Proteobacteria, Cyanobacteria, Bacteroidota, and Actinobacteriota are the most abundant bacterial phyla in these sauerkrauts. They also analyzed the amino acid, organic compounds and other metabolite content in these products. The analyses of the study are well conducted. However, the main criticism of the current manuscript is its data presentation. For most figures, the fonts in the panels are illegible and overlapping, making it difficult to understand. The figures lack the appropriate panel labels. As a result, it is difficult for the readers to often correlate the legend with the corresponding panels. The manuscript will benefit from well labelled figure panels with detailed figure legends.

Reviewer 3 Report

The manuscript is an interesting and well-written work focused on microbial and metabolite profiling of Chinese sauerkraut with and without microorganism addition.

Few minor corrections:

- In Title, change 'diversities' to 'diversity'

- In Abstract, define the abbreviations in line 2. Do the same in Tables 1 and 2.

- In Abstract, briefly mention which techniques were used

- In Abstract, is it cystine or cysteine?

- In Introduction, lines 6 and 7, yeasts and molds do not need to be in italics

- Section 2.1, explain what LLS, HX, QM and ZLL are.

- Section 2.3, describe the sequencing method and equipment

- Sections 2.5 and 2.6 can be merged

- Table 1, L. paracasei, L. acidophilus

- Discussion, are your results in agreement with these from similar studies?

- Discussion, are your finding relevant to sauerkraut products from other countries?

Round 2

Reviewer 1 Report

The manuscript has been well rewritten and appears easier to understand and read.
The introduction has been rewritten, and the various parts are more coherent and in line with the purpose of the paper. The results and conclusions are more clearly stated and adequately commented.

Author Response

Thank you very much for your valuable suggestions. Your suggestions have improved the quality of the whole manuscript.